# Molecular Parameters of Tert-Butyl Chloride and Its Isotopologues Determined from High-Resolution Rotational Spectroscopy

**Chao Jiao, Sheng-wen Duan, Yi Wu, Ming Sun \*, Qian Chen \*, Pei-yu Fang and Da-peng Wang**

School of Electronic and Optical Engineering, Nanjing University of Science and Technology, Nanjing 210094, China; cjiao@njust.edu.cn (C.J.); 117104021762@njust.edu.cn (S.-w.D.); 118104010120@njust.edu.cn (Y.W.); 118104021883@njust.edu.cn (P.-y.F.); dpWang@njust.edu.cn (D.-p.W.)

\* Correspondence: msun@njust.edu.cn (M.S.); chenq@njust.edu.cn (Q.C.)

**Abstract:** A broadband chirped-pulse Fourier transform microwave spectrometer was used to detect the rotational spectra of the products of a chemical reaction in the gas phase from 1-18 GHz under the supersonic expansion condition. In natural abundance, pure rotational energy level transitions of tert-butyl chloride and its isotopologues ($^{13}C$, $^{37}Cl$) were observed and assigned. The rotational spectral parameters (rotational constant, quadrupole coupling constant and centrifugal distortion constant) of these isotopologues were determined. The experimental results are in great agreement with the calculated values of quantum chemistry and the spectral parameters in the literature. The accuracy and the capability for chemical detection of our homemade rotational spectrometer were verified by this experiment.

**Keywords:** rotational spectroscopy; isotopologue; microwave spectrometer; tert-butyl chloride

---

## 1. Introduction

In the field of molecular rotational spectroscopy, high-resolution laboratory experiments are mostly combined with quantum chemical calculations to facilitate spectroscopic assignments for researchers engaged in chemical detection [1–5]. The majority of molecules have a unique set of rotational spectra in the microwave to terahertz band, i.e., the so-called molecular fingerprints. On the basis of quantum chemical calculation, molecular rotational spectroscopy can fit extremely accurate three-dimensional structures of free molecules and describe the local electric field gradient distribution caused by electron arrangement [6,7]. Therefore, it is quite important to capture pure rotational high-precision energy level transitions of molecules. Generally, the Fourier transform detection technology is used to quickly capture the hyperfine rotational spectra of gaseous substances. At present, two types of Fourier transform spectrometers are mainly used in the microwave band, one is the narrowband microwave spectrometer based on the Fabry−Perot cavity, while the other is the broadband microwave spectrometer based on chirped-pulse linear frequency modulation. The narrowband microwave spectrometer was designed and built by Professor Flygare of the University of Illinois in the early 1980s [8], with high sensitivity and high resolution. The broadband microwave spectrometer was successfully developed by the Pate's team at the University of Virginia in 2008 [9]. Compared with the narrow-band microwave spectrometer, the single scanning bandwidth of the broadband type can be improved by several orders of magnitude, but with lower resolution and sensitivity. Nowadays, many laboratories use both broadband and narrowband microwave spectrometers to improve molecular detection efficiency. This combination means that firstly samples are quickly scanned by a broadband microwave spectrometer, and then are rescanned by a narrowband microwave spectrometer in specific frequency regions with high resolution.

In recent years, with the continuous updating of microwave electronic devices [10–12], Fourier transform microwave spectroscopy detection technology has developed rapidly. Researchers not only improved the scanning bandwidth, but also took the high sensitivity and high resolution into account at the same time, and provided a series of auxiliary sample preparation technologies for spectrometers, such as laser photolysis [13] and high voltage discharge [14]. In addition to dealing with problems such as molecular structure and intermolecular interactions [15,16], FTMW spectroscopy is also applied to study the dynamic process of complex chemical reaction systems [17], and sensitively detect chiral compounds [18]. Therefore, it plays a significant role in the fields of chemical analysis, pharmaceutical detection and radio astronomy [19,20].

Tert-butyl chloride is a symmetric top molecule with a high degree of symmetry ($C_{3v}$ symmetry). The existence of a large number of similar trimethyl compounds has made relevant studies attractive [21]. Due to the nuclear coupling of halogen and the torsional oscillation or internal rotation of methyl groups, their microwave spectra become complex. In 1950, Williams and Gordon first studied the millimeter wave rotational spectra of tert-butyl bromide, tert-butyl chloride and tert-butyl iodine, and obtained their rotational constants, moments of inertia and carbon-halide bond lengths [22]. Subsequently, Gierszal and Legon successively analyzed the hyperfine structure of nuclear quadrupole coupling of $(CH_3)_3C^{35,37}Cl$, and gave the quadrupole coupling constants of chlorine [23,24]. What is more, Kassi and her coworkers also analyzed the rotational spectra of tert-butyl chloride, tert-butyl bromide and their isotopologues ($^{13}C$, $D_3$ full deuterium) and redetermined the bond length of C-X (x = Cl, Br) [25]. So far, the complete rotational spectra from J=1←0 to J=4←3 for tert-butyl chloride and its isotopologues ($^{13}C$, $^{37}Cl$) have not been measured, especially the low-order rotational transitions.

Therefore, we utilized a chirped-pulse Fourier transform microwave (cp-FTMW) spectrometer to measure the pure rotational energy level transitions of tert-butyl chloride and its isotopologues ($^{13}C$, $^{37}Cl$) in the 1–18 GHz frequency band. By fitting and analyzing the measured transitions, the rotational constants, quadrupole coupling constants and centrifugal distortion constants are well determined for each isotopologue.

## 2. Experiment

### 2.1. Experimental Instrument

The rotational spectra of tert-butyl chloride and its isotopologues ($^{13}C$, $^{37}Cl$) were measured by a broadband cp-FTMW spectrometer built at Nanjing University of Science and Technology [26]. One single scan bandwidth of the spectrometer is 2 GHz in the working frequency range of 1–20 GHz. Its working principle is consistent with the design of current mainstream broadband microwave spectrometers, as shown in Figure 1. First of all, the sample gas passes through the solenoid valve nozzle at the injection end and enters the vacuum chamber by ultrasonic expansion. When the gas beam reaches the center of the sample chamber after a one millisecond delay, a fast linear microwave frequency sweep of 2 GHz bandwidth, that is upconverted by mixing a chirped pulse of 1 GHz bandwidth from the arbitrary waveform generator (AWG, 2.5 G·sample/s) with a single frequency from the microwave source (1–20 GHz), radiates from a double-ridged horn antenna to excite molecules in the beam. Due to the existence of a spherical aluminum mirror, the broadband microwave pulse can be reflected back once to enhance the excitement. After that, the molecular relaxation emission signal (FID, free induction decay) is received by the same feedhorn and transferred out of the chamber through the circulator. The FID signal is then amplified and downconverted to fundamental frequency by mixing with the same single frequency from the microwave source before it is collected by a high-speed digital oscilloscope (40 G·sample/s). The AWG, microwave source and digital oscilloscope are all locked by a 10 MHz rubidium frequency standard oscillator to ensure phase matching and stability during signal transmission. The single-pole single-throw (SPST) electronic switch is used to protect downstream electronic devices from high-power radiation damage.

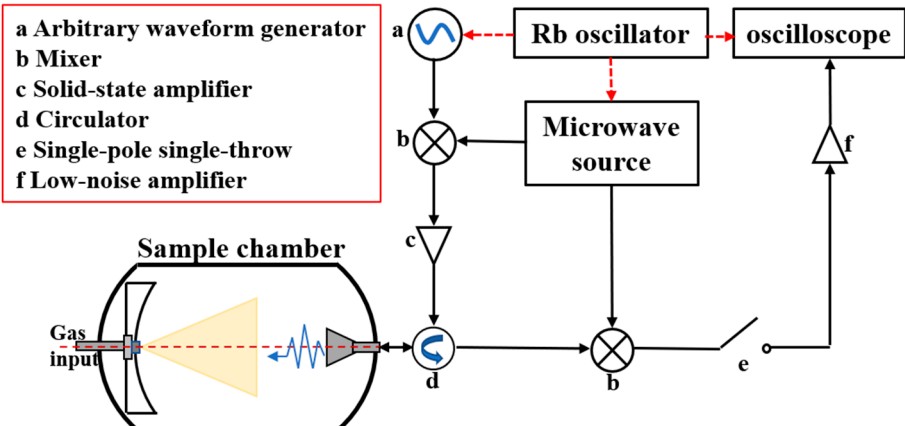

**Figure 1.** Schematic diagram of broadband cp-FTMW spectrometer.

In order to improve the sensitivity of the instrument, special design changes have been made in this work compared with the mainstream broadband microwave spectrometer. (1) A reflective focusing spherical aluminum mirror was designed to compensate for the lack of excitation microwave power; (2) the vacuum chamber, the reflective aluminum mirror and the gas injection nozzle were arranged in a coaxial manner to enhance the interaction between the molecular beam and the excitation radiation, so as to improve the excitation efficiency; (3) the broadband technology of multiple free induction decay (multiple FID) was developed to improve the signal sampling efficiency.

*2.2. Experimental Method and Theoretical Calculation*

Tert-butyl chloride and its isotopologues are the main products of a chemical reaction between tert-butyl alcohol and hydrogen chloride. The products were formed by flowing a mixed carrier gas (0.5% HCl + 99.5% Ar/V) through a U-tube with tert-butyl alcohol (99%, purchased from Sigma-Aldrich) inside at room temperature. At 60 psi back pressure, the sample gas was supersonic expanded into the vacuum chamber through the solenoid valve nozzle (general valve series 9, diameter 0.9 mm) to bring the rotational temperature of the sample molecules down to 1–10 Kelvin, so as to optimize the molecular population on the rotational energy levels. The gas pulse frequency was 1 Hz, and the jet duration was 500 μs. After each injection, the oscilloscope collected 23 FID signals and cut them evenly and average, that was our newly developed broadband technology of multiple free induction decay. In the final spectrum, 10,000–20,000 FID signals were averaged to obtain better spectral signal-to-noise ratio compared with the previous work in [26]. Thus, rotational transitions of isotopologues ($^{13}$C, $^{37}$Cl) could be detected in their natural abundance.

The ab initio electronic structure of tert-butyl chloride, as shown in Table 1, was calculated and optimized by Gaussian03 software at the MP2/6-311++G (D, P) theory level [27] to provide structural information (three-dimensional coordinates, bond length, bond angle) and rotational constants. Figure 2 is the calculated three-dimensional structure diagram and plane projection diagram of tert-butyl chloride. It is obvious that the molecule is highly symmetrical. The moment of inertia and rotational angular momentum of tert-butyl chloride and its $^{13}$C, $^{37}$Cl substituted isotopologues could be predicted by using PMIFST (Principal Moments of Inertia from Structure) [28] combined with Gaussian03 calculated three-dimensional coordinate matrix, to guide our initial spectroscopic assignment and fitting procedures.

**Table 1.** The rotational constant, moment of inertia and rotational angular momentum of $(CH_3)_3C^{35}Cl$, $(CH_3)_3C^{37}Cl$, $(CH_3)_3{}^{13}C^{35}Cl$, $(CH_3)_3{}^{13}C^{37}Cl$ predicted by Gaussian03 [27] and PMIFST [28].

| Spectral Parameter | Tert-Butyl Chloride and Its Isotopologues | | | |
| --- | --- | --- | --- | --- |
| | $(CH_3)_3C^{35}Cl$ [d] | $(CH_3)_3C^{37}Cl$ [d] | $(CH_3)_3{}^{13}C^{35}Cl$ | $(CH_3)_3{}^{13}C^{37}Cl$ |
| $A$(MHz) [a] | 4557.36740 | 4557.36740 | 4571.70485 | 4571.70485 |
| $B$(MHz) [a] | 3035.35346 | 2970.78357 | 3025.45983 | 2960.51347 |
| $C$(MHz) [a] | 3035.35068 | 2970.78091 | 3025.45734 | 2960.51109 |
| $I.\,a$(amu·Å$^2$) [b] | 110.892750 | 110.892750 | 110.544977 | 110.544977 |
| $I.\,b$(amu·Å$^2$) [b] | 166.497580 | 170.116399 | 167.042048 | 170.706538 |
| $I.\,c$(amu·Å$^2$) [b] | 166.497732 | 170.116552 | 167.042185 | 170.706575 |
| $P.\,a$(amu·Å$^2$/s) [c] | 111.051277 | 114.670097 | 111.769630 | 115.434120 |
| $P.\,b$(amu·Å$^2$/s) [c] | 55.446453 | 55.446453 | 55.272556 | 55.272556 |
| $P.\,c$(amu·Å$^2$/s) [c] | 55.446301 | 55.446301 | 55.272419 | 55.272419 |

[a] Rotational constant; [b] moment of inertia; [c] rotational angular momentum; [d] in [26].

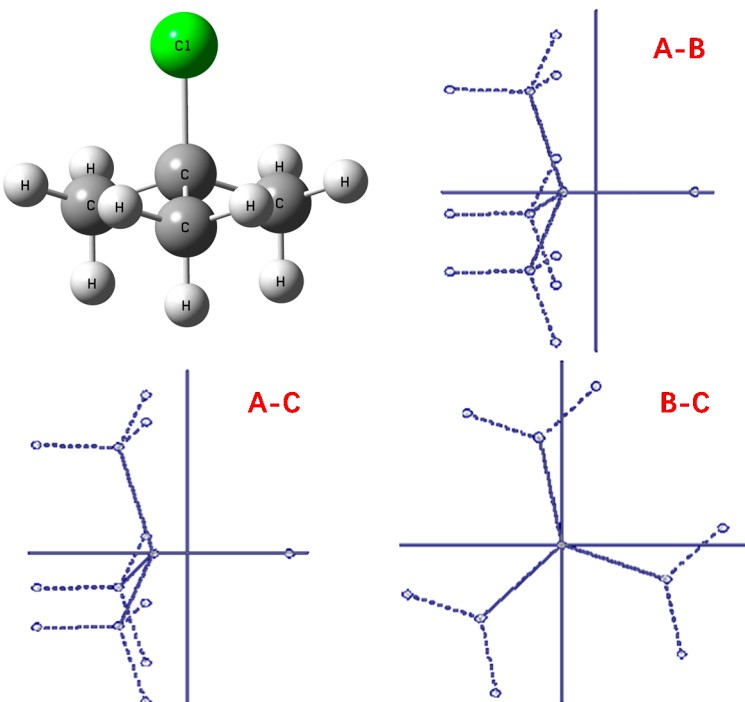

**Figure 2.** Three-dimensional structure diagram (upper left) and plane projection (A-B, A-C, B-C) of tert-butyl chloride calculated by Gaussian03 software at the MP2/6-311++G (D, P) theory level.

## 3. Results

The pure rotational spectra of tert-butyl chloride and its isotopologues in natural abundance were measured by cp-FTMW spectrometer in the frequency range of 1–18 GHz. The quanta $J = 1\leftarrow 0$ to $J = 3\leftarrow 2$ rotational energy level transitions of four substances ($(CH_3)_3C^{35}Cl$, $(CH_3)_3C^{37}Cl$, $(CH_3)_3{}^{13}C^{35}Cl$, $(CH_3)_3{}^{13}C^{37}Cl$ have been observed, as shown in Figure 3. Figure 4 displays the hyperfine splitting spectra of $J = 1\leftarrow 0$ transitions arising from Cl nuclei of each species. By using Pickett's SPCAT/SPFIT (spectral fitting analysis software) [29], the observed rotational spectral lines were assigned and high precision spectral parameters (including rotational constant $A$, $B$, $C$, centrifugal distortion constant $D_J$, $D_{JK}$ and quadrupole coupling constant $eQq$) of each species were well determined, as shown in Table 2.

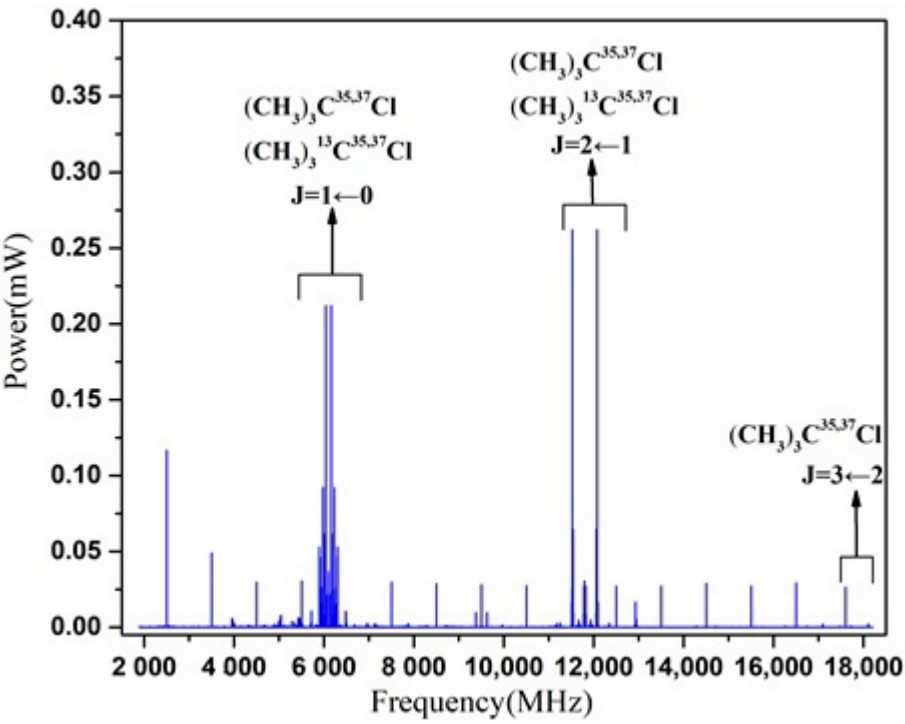

**Figure 3.** Diagram of the $J = 1 \leftarrow 0$ to $J = 3 \leftarrow 2$ rotational transitions of $(CH_3)_3C^{35}Cl$, $(CH_3)_3C^{37}Cl$, $(CH_3)_3{}^{13}C^{35}Cl$, $(CH_3)_3{}^{13}C^{37}Cl$ detected in the 1–18 GHz frequency region.

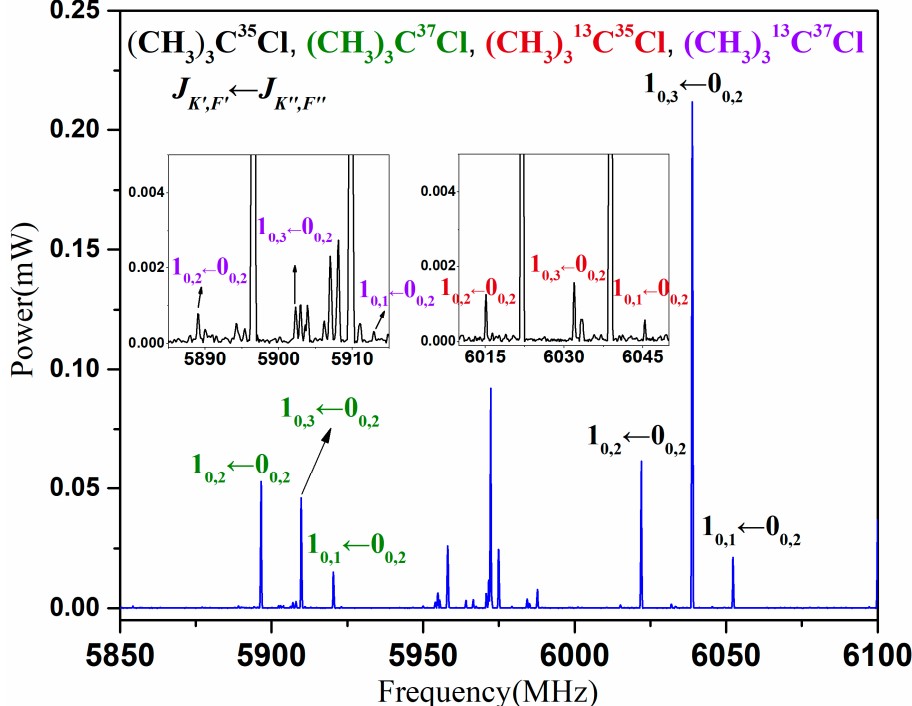

**Figure 4.** Diagram of measured hyperfine splitting of ground state ($J = 1 \leftarrow 0$) energy level transitions ($J'_{K'F'} \leftarrow J''_{K''F'}$) of $(CH_3)_3C^{35}Cl$, $(CH_3)_3C^{37}Cl$, $(CH_3)_3{}^{13}C^{35}Cl$, $(CH_3)_3{}^{13}C^{37}Cl$.

**Table 2.** Spectral parameters of $(CH_3)_3C^{35,37}Cl$ and $(CH_3)_3{}^{13}C^{35,37}Cl$ obtained in this work [a] and relevant values in the reference.

| Spectral Parameter | Tert-Butyl Chloride and Its Isotopologues | | | |
|---|---|---|---|---|
| | $(CH_3)_3C^{35}Cl$ | $(CH_3)_3C^{37}Cl$ | $(CH_3)_3{}^{13}C^{35}Cl$ | $(CH_3)_3{}^{13}C^{37}Cl$ |
| $B$(MHz) | 3017.721862(66) | 2953.570814(67) | 3014.284148(62) | 2949.821282(65) |
| | 3017.7177(9) [g] | 2953.5717(8) [g] | 3014.285415(40) [f] | 2949.8215(62) [f] |
| $D_J$(kHz) [b] | 0.9675(50) | 0.3808(52) | −1.6551(42) | −0.4812(40) |
| | 0.6(1) [g] | 0.6(l) [g] | 0.518(27) [f] | 0.511(35) [f] |
| $D_{JK}$(kHz) [b] | −0.2807(153) | −0.8866(162) | 0.01313(128) | −0.7978(100) |
| | 1.2(3) [g] | 1.2(3) [g] | 1.158(73) [f] | 1.246(26) [f] |
| $eQq$(MHz) [c] | −67.25407(104) | −53.10206(107) | −67.33081(93) | −53.07129(96) |
| | −67.312(3) [g] | −53.053(3) [g] | −67.3266(41) [f] | −53.0694(64) [f] |
| $RMS$(MHz) [d] | 0.0021 | 0.002369 | 0.001825 | 0.002044 |
| $N$ [e] | 24 | 22 | 30 | 31 |

[a] values without superscript are from this work; [b] centrifugal distortion constant; [c] quadrupole coupling constant; [d] the root-mean-square error of observed transition frequencies; [e] number of fitted transitions; [f] in [24]; [g] in [25].

## 4. Analysis and Discussion

In this experiment, the pure rotational spectra of tert-butyl chloride and its isotopologues ($^{13}C$, $^{37}Cl$) were studied in the range of 1–18 GHz by cp-FTMW spectrometer and quantum chemical calculation. From the calculation results, it can be found that the rotational constants $A > B = C$, and moments of inertia $I.\ a < I.\ b = I.\ c$, which clearly confirm that these molecules belong to prolate symmetric top. As shown in Table 2, a total of 107 transition lines were observed and assigned, of which 24 for $(CH_3)_3C^{35}Cl$, 22 for $(CH_3)_3C^{37}Cl$, 30 for $(CH_3)_3{}^{13}C^{35}Cl$ and 31 for $(CH_3)_3{}^{13}C^{37}Cl$. The root-mean-square errors of fitted transitions are no more than 2.5 kHz, indicating that the transition frequencies measured by the experiment are in good agreement with the predicted frequencies. As shown in Figures 2 and 3, after 20,000 times of averaging, the maximal signal-to-noise ratio of the spectrum was up to 2000, so that the low-order rotational transitions of each species were completely captured.

In this work, we measured the rotational energy level transition of $(CH_3)_3C^{35,37}Cl$, $(CH_3)_3{}^{13}C^{35,37}Cl$ in the low frequency band and obtained their exact rotational parameters by fitting and analyzing the rotational transitions, as shown in Table 2. Some data of $(CH_3)_3C^{35,37}Cl$ have been published in previous work. Due to the accurate measurement of the low frequency part, the calculation of the quadrupole coupling constants of isotopologues is better. The measurement results of rotational constants and quadrupole coupling constants are accurate, and their measurement accuracy can reach about 0.002%. The standard deviation of a single parameter is about five times less than that in the literature [24], which is helpful to provide a more accurate parameter standard for the quantum chemical calculation of the molecular structure. This is owing to improvements in the hardware design and new signal acquisition technology applied for our spectrometer. However, the measurement accuracy of centrifugal distortion is about 0.02%, but still better than the results in the literature [24,25], although the lack of millimeter-wave spectral data could result in inaccurate fitting. It can be found that, compared to the central carbon isotopes ($^{12}C$ and $^{13}C$), halogen isotopes ($^{35}Cl$ and $^{37}Cl$) can result in bigger difference for both rotational and quadrupole coupling constants. $(CH_3)_3C^{35}Cl$ and $(CH_3)_3{}^{13}C^{35}Cl$ with the same $^{35}Cl$ isotope have close rotational constants and almost equal quadrupole coupling constants. Similar conclusion can be made for $(CH_3)_3C^{37}Cl$ and $(CH_3)_3{}^{13}C^{37}Cl$. When comparing $(CH_3)_3{}^{13}C^{35}Cl$ and $(CH_3)_3{}^{13}C^{37}Cl$, the rotational and quadrupole coupling constants are off about 64.462866 MHz and -14.25952 MHz respectively. For the rotational constants, the halogen isotope can make a difference mainly due to its far off the molecule's center of mass. For the quadrupole constants, charge distribution in the halogen nuclei results in the very different spectroscopic hyperfine splitting between $(CH_3)_3{}^{13}C^{35}Cl$ and $(CH_3)_3{}^{13}C^{37}Cl$.

## 5. Conclusions

In this experiment, the rotational spectra of tert-butyl chloride and its $^{13}$C, $^{37}$Cl substituted isotopologues were all detected with high resolution in the range of 1–18 GHz, which demonstrates the sensitivity of cp-FTMW spectrometer to trace substances. The exact rotational constants and quadrupole coupling constants were well determined, that could be used as the numerical basis for the quantum computation of structure and bonding issues. From a spectroscopic view, the tert-butyl chloride with three identical methyl groups and different halogen isotopes can also act as a chemical model for the research in both torsional oscillation [30–32] and the hyperfine coupling effect of halogen nuclei [33,34] as well.

**Author Contributions:** C.J., S.-w.D., Y.W., P.-y.F., D.-p.W. designed and performed the experiments. C.J. analyzed the data and wrote the article. M.S. and Q.C. reviewed and edited the manuscript. C.J. responded to the reviews and revised the manuscript. All authors have read and agreed to the published version of the manuscript.

**Funding:** This research was funded by the National Natural Science Foundation of China (61627802, U1531107); and the Open Project Program of Jiangsu Key Laboratory of Spectral Imaging & Intelligent Sense (3091801410401).

**Acknowledgments:** The authors would like to thank the editors and the reviewers for their comments on an earlier draft of this article.

**Conflicts of Interest:** The authors declare no conflict of interest.

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
