# Peer review of "Molecular Parameters of Tert-Butyl Chloride and Its Isotopologues Determined from High-Resolution Rotational Spectroscopy"

_applsci, doi:10.3390/app10217650_

Round 1

Reviewer 1 Report

Review report

The Authors present a new experimental determination of molecular parameters for some isotopologues of (CH3)3CCl, obtained by analyzing their microwave spectra. The experimental results are complemented with quantum chemistry calculations.

In its present state I don’t find the manuscript suitable for publication because (a) I am not convinced that there is enough new scientific result which is worth publishing and because (b) the quality of the presentation is below that which is suitable for a scientific journal in my opinion.

If the Authors carry out a major revision, addressing the points raised below, the paper might be suitable for publication.

(1) The title of the paper is misleading. Regarding the scientific results presented, which are molecular parameters determined from rotational spectra, it is not utilized at any point how the measured species were created. The chemical reaction used to generate the sample is not investigated in any sense, it should not appear in the title in my opinion. I also find the expression “isotopologue sensing” misleading. In my understanding, “sensing” indicates that trace amounts of chemicals are detected during some process, which is monitored. In this work it seems to me that abundant amount of chemicals were created and their spectra recorded. A title along the line “Molecular parameters of tert-butyl chloride and its isotopologues determined from high-resolution rotational spectroscopy” would reflect the contents of the paper much better in my opinion.

(2) The paper starts with the sentence “Laboratory molecular rotational spectroscopy combined with quantum chemical calculation is a new research direction of molecular spectroscopy” , which I believe is simply not true. Laboratory rotational spectroscopy has been around for many decades, for example there is already is a review on the subject from as early as 1948. [W. Gordy, Rev. Mod. Phys. 20, 668–717, (1948)]. And rotational spectroscopy was certainly assisted from the earliest days of quantum chemistry, see for example [K. Yamanouchi et al. J. Mol. Struct. 126, 321-330 (1985)]. I find it disturbing that only two self citations appear on this very broad topic. Such a lack of proper coverage of previous literature is not acceptible in my opinion.

(3) One of my main concerns is that by reading the manuscript it is not clear to me what exactly are the novel results, and whether a scientific contribution worth publishing was made.
(3.1) In lines 168-169, the Authors write “and the standard deviation of single parameter is about 5 times less than that in the literature”. This is a clear message, however, no citation is given to the previous works to which the comparison is made.
(3.2) Table 2 seems to present most of the determined molecular parameters. Most of the numbers, however, are indicated to be taken from previous works and share the same accuracy as the numbers which seem to be determined in this work. Please clarify what are the numbers determined in this work and how do they compare to previous results.
(3.3) How was it possible to obtain the claimed high accuracy? It is not clear to me whether the experimental setup is the same as in Ref. 2 of the paper, or were there any improvements? This should be clarified.

By reading the current form of the manuscript, I can only acknowledge as a new result that a few new molecular parameters were determined with an accuracy higher than previously available. I am not convinced that the numerical value of a few molecular parameters, none of which seems particularly interesting for any specific reason, is worth publishing. Why would this be interesting to the community? The Author’s claim that the new parameters can be used as benchmark for testing theoretical methods is not convincing in my opinion.

On the other hand, if a new and useful experimental setup has been developed, or the new molecular parameters are valuable for the community for some reason, then the work is worth publishing, but in these cases these need to be emphasized.

(4) Abstract, line 10: The Authors write “rotational spectrum of a chemical reaction”. I believe a chemical reaction has no rotational spectrum. Chemical components have rotational spectra.

(5) The theoretical calculations were carried out on the MP2/6-311++G(d,p) level. This is in fact a rather low quality theoretical approach, and not surprisingly the agreement with experiment is not outstanding, only 2-3 digits. The high accuracy experimental work could be better supported with higher quality calculations.

(6) Lines 122-125: The Gaussian calculations provide not only geometrical structures but the rotational constants and other parameters as well. Why was the PMFIST software used to obtain the molecular parameters? Also, citation for the PMFIST software is missing.

(7) In the footnote of Table 2: What quantity does the RMS stand for?

(8) Line 157: The Authors write “long symmetric top”. I believe the correct expression is prolate symmetric top.

(9) Line 165: Table 3 is referenced, but there is no table 3 in the paper.

(10) Line 166: The Authors write “which is slightly different from the data recorded in the literature”. What does slightly different mean? Please be more specific. Again, there is no citation given to the literature the Authors are referring to.

Author Response

Response to Reviewer comments

The Authors present a new experimental determination of molecular parameters for some isotopologues of (CH3)3CCl, obtained by analyzing their microwave spectra. The experimental results are complemented with quantum chemistry calculations.

In its present state I don’t find the manuscript suitable for publication because (a) I am not convinced that there is enough new scientific result which is worth publishing and because (b) the quality of the presentation is below that which is suitable for a scientific journal in my opinion.

If the Authors carry out a major revision, addressing the points raised below, the paper might be suitable for publication.

Point 1:

The title of the paper is misleading. Regarding the scientific results presented, which are molecular parameters determined from rotational spectra, it is not utilized at any point how the measured species were created. The chemical reaction used to generate the sample is not investigated in any sense, it should not appear in the title in my opinion. I also find the expression “isotopologue sensing” misleading. In my understanding, “sensing” indicates that trace amounts of chemicals are detected during some process, which is monitored. In this work it seems to me that abundant amount of chemicals were created and their spectra recorded. A title along the line “Molecular parameters of tert-butyl chloride and its isotopologues determined from high-resolution rotational spectroscopy” would reflect the contents of the paper much better in my opinion.

Response 1:

Thank you for the revision advice. We noticed this misleading part and changed the title to “Molecular parameters of tert-butyl chloride and its isotopologues determined from high-resolution rotational spectroscopy”, that is a more appropriate title.

Point 2:

The paper starts with the sentence “Laboratory molecular rotational spectroscopy combined with quantum chemical calculation is a new research direction of molecular spectroscopy”, which I believe is simply not true. Laboratory rotational spectroscopy has been around for many decades, for example there is already is a review on the subject from as early as 1948. [W. Gordy, Rev. Mod. Phys. 20, 668–717, (1948)]. And rotational spectroscopy was certainly assisted from the earliest days of quantum chemistry, see for example [K. Yamanouchi et al. J. Mol. Struct.126, 321-330 (1985)]. I find it disturbing that only two self citations appear on this very broad topic. Such a lack of proper coverage of previous literature is not acceptible in my opinion.

Response 2:

Thank you for the revision advice. We noticed the mistake and added more appropriate references to give better coverage in the introduction.

Point 3:

One of my main concerns is that by reading the manuscript it is not clear to me what exactly are the novel results, and whether a scientific contribution worth publishing was made.
(3.1) In lines 168-169, the Authors write “and the standard deviation of single parameter is about 5 times less than that in the literature”. This is a clear message, however, no citation is given to the previous works to which the comparison is made.
Response 3.1:

Thank you for the comments. We noticed this misleading part about the novel results. We stressed the results from this work in both the text and Table 2. We gave clear citation regarding the previous work for the comparison. We changed it to “The measurement accuracy of the rotational constants and quadrupole coupling constants can reach about 0.002%. The standard deviations of those parameters are normally about 5 times less than that in the literature [24]”. And in Table 2, the standard deviations of all the parameters from this work and literature are in the brackets for comparison.

(3.2) Table 2 seems to present most of the determined molecular parameters. Most of the numbers, however, are indicated to be taken from previous works and share the same accuracy as the numbers which seem to be determined in this work. Please clarify what are the numbers determined in this work and how do they compare to previous results.
Response 3.2:

Thank you for the revision advice. We noticed this misleading part and changed it to “In this work, we measured the rotational energy level transition of (CH3)3C35,37Cl, (CH3)313C35,37Cl in the low frequency band and obtained their exact rotational parameters by fitting and analyzing the rotational transitions, as shown in Table 2. Some data of (CH3)3C35,37Cl from our group have been published in previous work. Due to our accurate measurement of the low frequency lines, the quadrupole coupling constants of all isotopologues are better determined. The measurement accuracy of the rotational constants and quadrupole coupling constants can reach about 0.002%. The standard deviations of those parameters are normally about 5 times less than that in the literature [24], which is helpful to provide more concrete standards for the quantum chemical calculation of this molecule. However, the measurement accuracy of centrifugal distortion is about 0.02%, but still better than the results in the literature [24,25], although the lack of millimeter-wave spectral data could result in inaccurate fitting.”

(3.3) How was it possible to obtain the claimed high accuracy? It is not clear to me whether the experimental setup is the same as in Ref. 2 of the paper, or were there any improvements? This should be clarified.

Response 3.3:

Thank you for the revision advice. We noticed this misleading part and explained the reasons for the high resolution regarding the experimental setup in the article.

  1. Compared with other existing spectral instruments, there are several improvements in the specific design of the spectrometer compared with the mainstream broadband microwave spectrometer: “(1) a reflective focusing spherical aluminum mirror is designed to compensate for the lack of excitation power of the broadband microwave source; (2) the vacuum chamber, the reflective focusing spherical aluminum mirror and the gas injection nozzle are arranged in a coaxial manner to enhance the contact between the molecular beam and the excited microwave, so as to improve the excitation efficiency; (3) broadband multiple free induction decay (multiple FID) technology is developed to improve the signal sampling efficiency and the signal sampling rate of the spectra.”
  2. Compared with the spectral instrument in reference [26], there is not much change, but the bandwidth and working range of the single scan are improved. “One single scan bandwidth of the spectrometer was improved to 2 GHz in the working frequency range of 1-20 GHz.”
  3. In addition, there are some changes in experimental Settings. “After each injection, the oscilloscope can collect 23 FID signals and cut them evenly and average. In the final spectrum, for each frequency band, there were 10,000 to 20,000 FID signals averaged to obtain better spectral signal-to-noise ratio compared with the previous work in [26].”

Point 4:

Abstract, line 10: The Authors write “rotational spectrum of a chemical reaction”. I believe a chemical reaction has no rotational spectrum. Chemical components have rotational spectra.

Response 4:

Thank you for the comments. We noticed this mistake and changed it to “A broadband chirped-pulse Fourier transform microwave spectrometer was used to detect the rotational spectra of the products of a chemical reaction in the gas phase from 1-18 GHz under the supersonic expansion condition.”

Point 5:

The theoretical calculations were carried out on the MP2/6-311++G(d,p) level. This is in fact a rather low quality theoretical approach, and not surprisingly the agreement with experiment is not outstanding, only 2-3 digits. The high accuracy experimental work could be better supported with higher quality calculations.

Response 5:

Thank you for the comments. Generally, for substances or unknown substances that have not been studied in rotational spectroscopy, more high-precision theoretical calculations are indeed required as the basis and guidance for rotational spectroscopy experiments. Here, we did not use higher-precision calculations in this experiment, because tertiary butyl chloride has already been calculated and part of its rotational spectrum has been studied. Therefore, we did not use the calculation with higher accuracy in this experiment, but applied the theoretical calculations carried out on the MP2/6-311++G(d,p) level, and referenced partial rotational constants in the references. On this basis, more accurate spectral parameters were easily obtained by our measurement of the rotational spectral lines and fitting analysis.

Point 6:

Lines 122-125: The Gaussian calculations provide not only geometrical structures but the rotational constants and other parameters as well. Why was the PMFIST software used to obtain the molecular parameters? Also, citation for the PMFIST software is missing.

Response 6:

Thank you for the revision advice. We noticed this mistake and changed it to “The ab initio electronic structure of tert-butyl chloride was calculated and optimized by Gaussian03 software at the MP2/6-311++G (D, P) theory level [27], and the structural information (three-dimensional coordinates, bond length, bond angle) of tert-butyl chloride and rotational constants were obtained. Fig. 2 is the three-dimensional structure diagram and plane projection diagram of tert-butyl chloride calculated by Gaussian03 software. It is obvious that the molecule is highly symmetrical. The moment of inertia and rotational angular momentum of tert-butyl chloride and its 13C, 37Cl substituted isotopologues can be predicted by using PMIFST (Principal Moments of Inertia From Structure) [28] combined with Gaussian03 calculated three-dimensional coordinate matrix.”

Point 7:

In the footnote of Table 2: What quantity does the RMS stand for?

Response 7:

Thank you for the revision advice. We noticed this mistake and explained that RMS means the root-mean-square error of observed transition frequencies.

Point 8:

Line 157: The Authors write “long symmetric top”. I believe the correct expression is prolate symmetric top.

Response 8:

Thank you for the revision advice. We noticed this mistake and changed it to “prolate symmetric top

Point 9:

 Line 165: Table 3 is referenced, but there is no table 3 in the paper.

Response 9:

Thank you for the revision advice. We noticed this mistake and changed it to Table 2.

Point 10:

Line 166: The Authors write “which is slightly different from the data recorded in the literature”. What does slightly different mean? Please be more specific. Again, there is no citation given to the literature the Authors are referring to.

Response 10:

Thank you for the revision advice. We noticed this mistake and changed it to “However, the measurement accuracy of centrifugal distortion is about 0.02%, but still better than the results in the literature [24,25], although the lack of millimeter-wave spectral data could result in inaccurate fitting.”

Reviewer 2 Report

This is a very nice paper on microwave specroscopy of tert-butyl chloride and its isotopomers, supported by quantum chemical calculations.  The experimental apparatus is an advanced chirp pulse spectrometer and the results are well presented and interpreted.

My main query to the authors is that the abstract does not seem to fit with this paper.  In the abstract, the authors state that they demonsrate the technique for a chemical reaction, but I could find no narrative about a chemical reaction in the paper.  While it is true that the molecule studied has been produced through a reaction, the reaction does not seem to have been studied. This must be clarified, and the abstract rewritten to more closely reflect the work presented.  If the authors believe they have studied the reaction, they should return to what they learnt about the reaction in the results and conclusion.  

Aside from that, I have a small number of minor comments for the authors:

  1. Figure 2 figure caption.  I would remove reference to the software from here, but put in the level of the calculations (e.g MP2/xxx)
  2. Figure 4 - the text is too small to read on the inset figures.
  3. The English could be improved in many places through the manuscipt, so I recommend that authors to pay for the editing service.
  4. It would be useful for the authors to add some references in the conclusions, namely to other work where torsional oscillations have been studied by this technique and its application to internal rotations of methyl groups.

Author Response

Response to Reviewer comments

This is a very nice paper on microwave specroscopy of tert-butyl chloride and its isotopomers, supported by quantum chemical calculations.  The experimental apparatus is an advanced chirp pulse spectrometer and the results are well presented and interpreted.

Point 1:

My main query to the authors is that the abstract does not seem to fit with this paper.  In the abstract, the authors state that they demonsrate the technique for a chemical reaction, but I could find no narrative about a chemical reaction in the paper.  While it is true that the molecule studied has been produced through a reaction, the reaction does not seem to have been studied. This must be clarified, and the abstract rewritten to more closely reflect the work presented.  If the authors believe they have studied the reaction, they should return to what they learnt about the reaction in the results and conclusion.  

Response 1:

Thank you for the revision advice. We noticed this misleading part and changed it to “A broadband chirped-pulse Fourier transform microwave spectrometer was used to detect the rotational spectra of the products of a chemical reaction in the gas phase from 1-18 GHz under the supersonic expansion condition.”

Point 2:

Figure 2 figure caption.  I would remove reference to the software from here, but put in the level of the calculations (e.g MP2/xxx)

Response 2:

Thank you for the revision advice. We made the change according to the comments.

Figure 2. Three-dimensional structure diagram (upper left) and plane projection (A-B, A-C, B-C) of tert-butyl chloride calculated by Gaussian03 software at the MP2/6-311++G (D, P) theory level.

Point 3:

Figure 4 - the text is too small to read on the inset figures.

Response 3:

Thank you for the revision advice. We readjusted the text size on the inset figures.

Figure 4. Diagram of measured hyperfine splitting of ground state (J=1¬0) energy level transitions (J’K’F’¬J’’K’’F’) of (CH3)3C35Cl, (CH3)3C37Cl, (CH3)313C35Cl, (CH3)313C37Cl.

Point 4:

The English could be improved in many places through the manuscipt, so I recommend that authors to pay for the editing service.

Response 4:

Thank you for the revision advice. We have noticed this problem and revised the whole article seriously.

Point 5:

It would be useful for the authors to add some references in the conclusions, namely to other work where torsional oscillations have been studied by this technique and its application to internal rotations of methyl groups.

Response 5:

Thank you for the comments. We have added references regarding torsional splitting in the conclusions.

Round 2

Reviewer 1 Report

Thank you for carefully considering the review comments, I believe the paper is now suitable for publication.